# Robust Preanalytical Performance of Soluble PD-1, PD-L1 and PD-L2 Assessed by Sensitive ELISAs in Blood

**DOI:** 10.3390/biomedicines10102534

**Published:** 2022-10-11

**Authors:** Kimberly Krueger, Zsuzsanna Mayer, Marc Kottmaier, Miriam Gerckens, Stefan Boeck, Peter Luppa, Stefan Holdenrieder

**Affiliations:** 1German Heart Center Munich, Clinics at the Technical University Munich, Institute of Laboratory Medicine, Munich Biomarker Research Center, 80636 Munich, Germany; 2German Heart Center Munich, Clinics at the Technical University Munich, Department of Cardiovascular Disease, 80636 Munich, Germany; 3Department of Internal Medicine III, University Hospital Munich-Grosshadern, 81377 Munich, Germany; 4Department of Clinical Chemistry, University Hospital of the Technical University Munich, 81675 Munich, Germany

**Keywords:** PD-1, PD-L1, PD-L2, preanalytical validation, ELISA, biomarker

## Abstract

The interaction between programmed death-1 receptor PD-1 and its ligands PD-L1 and PD-L2 is involved in self-tolerance, immune escape of cancer, cardiovascular diseases, and COVID-19. As blood-based protein markers they bear great potential to improve oncoimmunology research and monitoring of anti-cancer immunotherapy. A variety of preanalytical conditions were tested to assure high quality plasma sample measurements: (i) different time intervals and storage temperatures before and after blood centrifugation; (ii) fresh samples and repeated freeze–thaw-cycles; (iii) different conditions of sample preparation before measurement. Concerning short-term stability, acceptable recoveries for PD-1 between 80 and 120% were obtained when samples were kept up to 24 h at 4 and 25 °C before and after blood centrifugation. Similarly, recoveries for PD-L2 were acceptable for 24 h at 4 °C and 6 h at 25 °C before blood centrifugation and up to 24 h at 4 and 25 °C after centrifugation. Variations for PD-L1 were somewhat higher, however, at very low signal levels. Sample concentrations (ng/mL) were neither affected by the freezing process nor by repeated freeze–thaw cycles with coefficients of variation for PD-1: 9.1%, PD-L1 6.8%, and PD-L2 4.8%. All three biomarkers showed good stability regarding preanalytic conditions of sample handling enabling reliable and reproducible quantification in oncoimmunology research and clinical settings of anti-cancer immunotherapy.

## 1. Introduction

The interaction of programmed death-1 receptor (PD-1) with its ligands programmed death-ligand 1 (PD-L1) and 2 (PD-L2) represents a physiological mechanism to downregulate immune activity. The pathway is seen to be utilized by different tumors to evade immune response and elimination [1,2,3] and can be addressed by newly developed checkpoint inhibitors [4,5,6,7]. As the medication is costly and only 10–40% of patients benefit, there is a need for predictive markers to select and stratify responsive patients [6,8,9]. Currently, immunohistochemistry (IHC) methods based on anti-PD-L1 antibodies are mostly applied; however, they show inconsistent results depending on the specificity of the antibodies used. Recently, blood-based immunoassays were developed to sensitively detect and quantify soluble forms of PD-1, PD-L1, and PD-L2 [10]. This approach offers the advantage of quick, inexpensive, robust, and quantitative measurements in easily accessible patient blood that can be used to improve future oncoimmunology research as well as for the individual serial monitoring of the response to immune therapies in cancer patients [11,12,13]. The applicability and clinical value of soluble programmed death protein markers (PD markers) have already be shown by studies on patients suffering from diverse cancers, among them lung cancer, renal cell cancer, and melanoma receiving variable immune checkpoint inhibitor therapies [14,15].

Besides a thorough analytical validation [10], preanalytical handling conditions must be specified and standardized before the assays can be applied in clinical studies. It is well known that some protein profiles are affected by a variety of influencing factors occurring during collection and sample preparation processes [16,17]. The choice of sample collection tubes and time lags during transport to the laboratory before centrifugation as well as centrifugation speed differences can cause alterations. Different storage temperatures and the number of repeated freeze–thaw cycles can potentially also affect sample quality [18,19,20,21]. Hemolysis and platelet activation increase the concentrations of neuron-specific enolase (NSE) and lactate dehydrogenase (LDH), explaining the need for accurate sample handling and quick centrifugation after venipuncture [22,23,24]. Cytokines, such as interleukin-6 (IL-6), and also cholecalciferol (vitamin D_3_) are known to be unstable at room or fridge temperature, and have to be stored at −20 °C when measured after one (IL-6) or four days (vitamin D_3_) [25,26]. These examples underline the importance of assessing the influence of preanalytical factors when introducing new blood-based biomarkers in the diagnostic setting [27,28]. This becomes particularly relevant when changes in biomarker concentrations are considered by monitoring individual biomarker courses in serial measurements. Therefore, a variety of clinically relevant preanalytical conditions, including short-term stability, freeze effects, refreeze effects, and sample preparation before measurement, were tested on newly developed PD markers in the present study.

## 2. Materials and Methods

Measurements of the soluble biomarkers PD-1, PD-L1, and PD-L2 were performed with a self-developed enzyme-linked immunosorbent assay (ELISA). Two matching antibodies and recombinant protein standards were purchased from R&D Systems (Human PD-1 DuoSet^®^ ELISA: DY1086/Human PD-L1 DuoSet^®^ ELISA: DY156/Human PD-L2/B7-DC DuoSet^®^ ELISA: DY1224, R&D Systems, Inc., Minneapolis, MN, USA) to create the assay. Quantification was enabled using an eight-point standard curve from recombinant protein ranging from 0.0073 to 30 ng/mL for all three assays. The detection method was based on electrochemiluminescence which is captured by the Meso Quickplex SQ 120 analytical platform (Meso Scale Discovery, LLC., Rockville, MD, USA). Whole blood collection tubes used in the experiments included the matrices serum, lithium-heparin plasma, K3-ethylenediaminetetraacetic acid (K3-EDTA) plasma, and sodium-citrate plasma (S-Monovette^®^ 5.5 mL Z, S-Monovette^®^ 5.5 mL LH, S-Monovette^®^ 9 mL K3E, S-Monovette^®^ 5 mL 9NC; all: SARSTEDT AG&Co.KG, Nümbrecht, Germany). Blood samples were centrifuged with 3000 rcf at room temperature for 10 minutes to obtain serum or plasma. As intra-assay imprecision for all three biomarkers was very good (4.5–10%), single sample application experiments were carried out [6].

### 2.1. Standardization and Quality Control

Three quality control samples were prepared from pooled patient serum samples with low, medium, and high concentration of the corresponding biomarker. The concentrations for the pools named low, medium, and high were chosen in the lower, middle, and upper thirds of expected concentrations in human blood. Multiple aliquots with PD-1, PD-L1, and PD-L2 pools were stored at −80 °C to ensure sample stability. Controls were included in every plate to enable a plate performance comparison. Recoveries within 80–120% and coefficients of variation (CVs) of maximal 20% met acceptance criteria [29,30].

### 2.2. Short-Term Stability

The variables temperature, time, and centrifugation speed were examined in a short-term stability experiment comprising 23 different conditions (Figure 1). Three main experimental parts were executed separately: (i) storage of whole blood samples in the pre-centrifugation experiment, (ii) storage of plasma in the post-centrifugation experiment, and (iii) the comparison of three centrifugation speeds. Therefore, blood was collected from nine different participants and measured in single measurements to conduct all three parts of the experiment.

As the baseline control, one aliquot of plasma was stored immediately at −80 °C to serve as the reference concentration in the analysis. For the pre-centrifugation experiment (i), 14 whole blood aliquots were stored at 2–8 °C in the fridge, at room temperature (25 °C) or at 37 °C for 0, 3, 6, 24, 48, and 168 h in order to evaluate stress conditions during transport or shipping. Two further aliquots were part of the centrifugation experiment (iii) which compared the centrifugation velocities 1500, 3000 (standard), and 6000 rcf (Appendix A). The remaining whole blood was centrifuged (3000 rcf, 10 min, room temperature) to separate the plasma for the post-centrifugation experiments (ii). Plasma samples were then stored at 4 and 25 °C for 0, 3, 6, and 24 h. According to the experimental plan, all samples were frozen at −80 °C at their pre-specified time points until all samples were collected to be measured together on one plate.

### 2.3. Freeze Effects

Within 20 min after the blood draw, the samples were centrifuged (3000 rcf, 10 min, room temperature) and stored at 2–8 °C, room temperature, −20 °C and −80 °C for three hours before measurement. The aliquot stored at 2–8 °C was used as the reference sample in the analysis. These experiments were performed in serum and three different plasma types (heparin, EDTA, and citrate plasma). The blood of five participants was included in the pilot study in single measurements. All results were calculated as recoveries based on the fridge control.

### 2.4. Refreeze Effects

Samples were thawed once, twice, and three times and stored at room temperature for three hours to assure complete defrosting. The experiments were performed in serum and three different plasma types (heparin, EDTA, and citrate). In five participants, all four matrices were investigated in single measurements.

### 2.5. Sample Preparation before Measurement

This pilot approach investigated sample preparation immediately before plate application. It compared the mixing method (vortexing or pipetting) with unmixed samples as well as centrifuged and non-centrifuged plasma. The resulting six conditions were tested in heparin samples obtained from two participants, frozen at −80 °C for one week (Appendix A). Measurements were performed in duplicates. Two aliquots were stored for each participant, one in a standard sample tube and one in a cryotube (Eppendorf Safe-Lock Tube, 1.5 mL, Eppendorf QualityTM, Eppendorf AG, Hamburg, Germany; NuncTM CryoTubesTM, Thermo Fisher Scientific, Waltham, MA, USA). All samples were thawed motionless on the workbench. One aliquot was applied directly to the plate. The second aliquot was mixed by pipetting the plasma up and down several times. The third aliquot was mixed using the vortexer for 10 seconds, followed by plate application. All three aliquots were then centrifuged at a velocity of 3000 rcf for two minutes at room temperature and afterwards again pipetted on the plate. Mixing the samples with a pipette, followed by centrifugation (=C6) displayed the reference against which all other tested conditions were compared.

### 2.6. Ethics and Informed Consent

The sample collection was performed as part of quality control for biobanked blood samples for future study purposes with PD markers. Blood samples (maximum 25 mL) were taken from patients of the Department of Cardiology during routine venipuncture after patients had given their informed consent for blood collection for the Cardiovascular Biobank of the German Heart Centre Munich that is part of the Joint Biobank Munich and the German Biobank Node.

### 2.7. Statistics and Data Interpretation

Measurement data were analyzed using the Discovery Workbench 4.0.12 (LSR_4_0_12) software (Meso Scale Discovery, LLC., Rockville, MD, USA). Further statistical analysis graphs were drawn with Microsoft Excel 2010, including calculation of mean, standard deviation, recovery, and coefficient of variation. The requested range for recoveries was between 80% and 120%, whereas the coefficient of variation (CV) had to stay within ±20%. Graphically, the obtained results were depicted as bar or line charts. Acceptance criteria ranges were incorporated as dotted red lines. If possible, error bars indicated corresponding coefficients of variation.

## 3. Results

### 3.1. Short-Term Stability

Recoveries for centrifugation at 1500 rcf achieved mean recoveries and CVs of 97% (CV: 10%) for PD-1, 97% (CV: 13%) for PD-L1, and 109% (CV: 6%) for PD-L2 as compared with 3000 rcf. The results obtained for centrifugation at 6000 rcf were 102% (CV: 12%) for PD-1, 97% (CV: 16%) for PD-L1, and 107% (CV: 12%) for PD-L2 as compared with 3000 rcf (Figure 2 and Appendix A).

The calculated recoveries obtained in parts i and ii of the short-term stability experiment are listed in Table 1. PD-1 concentrations in the plasma sample met acceptance criteria with stable mean recoveries until 48 h for the storing temperatures of 4 and 25 °C. The mean recoveries at 37 °C presented with slightly higher variations throughout the whole observation period of 48 h. When stored after centrifugation in diverse conditions, PD-1 plasma concentrations revealed stable mean recoveries during the 24 h investigation time for both temperatures, 4 and 25 °C (Figure 3A,D and Appendix A, Table 1).

Similarly, the mean PD-L1 recoveries in whole blood were stable for 48 h for 4 and 25 °C. Mean concentrations measured at 37 °C were stable for six hours. After centrifugation, mean concentrations remained stable when samples were stored at 4 °C while there were somewhat higher variations if samples were kept at 25 °C (Figure 3B,E and Appendix A, Table 1).

Mean recoveries of PD-L2 concentrations in whole blood were stable at 4 and 25 °C at least up to 48 h while there were some variations at a storage temperature of 37 °C again. In summary, plasma could safely be stored at 4 and 25 °C at least for 24 h (Figure 3C,F and Appendix A, Table 1).

Regarding centrifugation speed (experiment part iii), there were no differences observed for alternative velocities of 1500 and 6000 rcf as compared with the reference centrifugation speed of 3000 rcf for all three analytes, PD-1, PD-L1, and PD-L2.

### 3.2. Freeze Effects

Comparing different blood matrices (serum, heparin-, EDTA-, and citrate-plasma), different storage temperatures (25, −20, and −80 °C) with the reference temperature of 4 °C revealed stable results over all materials, storage conditions, and for all three biomarkers: for PD-1, recoveries ranged from 99 to 101% at room temperature (25 °C), from 98 to 101% at −20 °C, and from 90 to 105% at −80 °C for all four materials (Figure 4A and Appendix A). Regarding PD-L1, recoveries ranged from 90 to 112% at room temperature, from 111 to 130% at −20 °C, and from 101 to 139% at −80 °C for all four materials. Highest mean recoveries were always obtained in citrate plasma (Figure 4B and Appendix A). For PD-L2, recoveries ranged from 95 to 103% at room temperature, from 99 to 108% at −20 °C, and from 102 to 115% at −80 °C for all four materials (Figure 4C and Appendix A). Corresponding coefficients of variation (CVs) for all three biomarkers are summarized in Appendix A.

Acceptance criteria of recoveries between 80% and 120% were met for all materials and temperatures investigated for PD-1 and PD-L2 as well as for PD-L1 in serum, heparin, and EDTA-plasma while citrate plasma yielded higher recoveries. For all features, error bars indicate the variation of single measurements.

### 3.3. Refreeze Effects

Performing up to three freeze–thaw cycles at −80 °C did not strongly influence the measurement results for all three analytes. For PD-1, serum, EDTA-, and citrate-plasma yielded mean recoveries in a range of 101–110% with CVs of 4–11%, and recoveries of heparin-plasma ranged within 80% and 120% though with a higher CV of 26–29% (Figure 5A and Appendix A). For PD-L1, mean recoveries of serum, heparin-, and EDTA-plasma ranged between 88% and 107% with CVs of 7–12% while citrate-plasma ranged at around 120% (Figure 5B and Appendix A). For PD-L2, mean recoveries ranged between 95% and 110% with CVs of 2–10% in all four tested materials (Figure 5C and Appendix A).

Acceptance criteria of recoveries between 80% and 120% were met for all materials and refreezing conditions and all three biomarkers. Only for PD-L1 in citrate-plasma, high recoveries were observed.

### 3.4. Sample Preparation before Measurement

Reconstitution of the sample immediately before measurement was investigated with respect to different types of sample mixing and centrifugation. For PD-1, mean recoveries obtained from the standard sample tube increased over 120% for conditions without centrifugation (C1–3) and no mixing but centrifugation (C4). Mean calculated recoveries in the cryotube were between 96 and 113% though conditions 1–3 showed high variations of 10–26%. Mean recoveries obtained from the standard sample tube for PD-L1 increased over 120% for conditions without centrifugation (C1–3). Mean calculated recoveries in the cryotube ranged between 93 and 116%. Regarding PD-L2, the mean recoveries for the first condition rose above 120% in both tested tubes. All other mean recoveries ranged between 98 and 117% though CVs of up to 25% were calculated (Appendix A).

In PD-1 only the conditions including a centrifugation step and only the cryotube showed acceptable recoveries. Regarding PD-L1, all conditions in the cryotube and the conditions C4 and C5 in the sample tube were in an acceptable range. The only acceptable condition for PD-L2 displayed C3 in the standard tube and C5 in the cryotube.

All obtained results were combined into a standard operating procedure (SOP) for future sample treatment (Table 2).

## 4. Discussion

The need to stratify patients for immune checkpoint inhibitor treatment requires reliable and informative biomarkers, based on analytical and preanalytical valid test systems. Standardized sample preparation and storage enabling highly comparable concentration measurements represent prerequisites for any biomarker assessment in clinical studies. Although a variety of investigations proved the necessity to specify preanalytical sample treatment, no general guideline exists so far. The guideline on ionized calcium from the Clinical and Laboratory Standards Institute (CLSI) was used for orientation purposes and adjusted to the needs of the investigated protein biomarkers PD-1, PD-L1, and PD-L2 [31]. The nature of proteins and feasibility resulted in the following four investigated categories: short-term stability, freeze effects, refreeze effects, and sample preparation before application. The obtained preanalytical data were analyzed and insights were summarized in an SOP.

All experiments proved a good stability of all three biomarkers PD-1, PD-L1, and PD-L2. Centrifugation speed in the range between 1500 and 6000 rcf had no influence on sample results for either of the biomarkers. Whole blood concentrations obtained in heparin blood collection tubes were stable at 4 and 25 °C, up to 48 h before processing. Up to 6 h of storage at 37 °C was acceptable, which could occur during shipment. Plasma concentrations remained stable for 24 h at 4 and 25 °C regarding PD-1 and PD-L2, whereas this only applies to 4 °C storage for PD-L1. All in all, blood collection tubes should be processed within three hours after venipuncture and temperatures over 30 °C should be avoided whenever possible. Obtained plasma must be measured or frozen within one hour. Adherence to this preanalytical procedure is important to avoid possible influences by hemolysis, platelet activation, or stressed cells that was challenged by the experimental plan. No impact of hemolysis is expected as PD markers are not present on the surface of erythrocytes, whereas the release of PD markers from stressed platelets could be relevant [32]. These issues were addressed by various means: longer time intervals and higher temperatures before centrifugation are supposed to stress cells and lead to higher hemolysis. Coagulation in serum tubes is associated with higher platelet activation as compared to plasma. By testing all these conditions in pre- and post-centrifugation experiments in detail, robust preanalytical conditions for sample handling were identified that enable safe procedures in routine laboratories.

No current assessment of PD-1, PD-L1, or PD-L2 protein stability was found in the literature. Its stability was compared with two protein groups known to be part of inflammatory and immunogenic processes, cytokines and immunogenic cell death markers, to be able to provide estimates. Compared to proinflammatory cytokines such as IL-6 or TNF-α, the protein death markers had a relatively long stability in unprocessed blood [33]. Whereas, the stability of the immunogenic cell death markers, hRAGE and HMGB1, was shown to be similar to that calculated for PD-1, PD-L1, and PD-L2 [34,35].

The freezing process provided stable results in serum and EDTA-plasma and showed higher variations in heparin and citrate plasma. Nevertheless, variations remained in the acceptable range and higher variations in PD-L1 concentrations were explainable by the naturally higher variation in low concentration samples. Conclusively, it can be stated that the freezing process did not affect the proteins in serum, heparin, or EDTA plasma, proving that the stability in repeated freeze–thaw cycles was important as study samples often undergo many different analysis procedures. It was shown that PD-1 is stable for three cycles in serum, EDTA, and citrate, whereas PD-L1 values did not change in serum, heparin, and EDTA. PD-L2 concentrations were not affected by repeated freeze and thaw cycles in any of the tested matrices. Samples should be mixed and centrifuged before plate application, preferably by pipetting. The use of cryotubes reduced the variation between replicates although the standard sample tubes also showed acceptable results in most preparation conditions.

Careful execution and many tested conditions allowed the estimation of the influence of a variety of potential interfering factors on the concentration of PD-1, PD-L1, and PD-L2. The findings were summarized in an SOP that will be applied to all future sample collections (Table 2).

The presented study faced some limitations. First, the results are only valid for the antibody clones and assay compositions investigated here and not for any other PD marker assay. As for all immunoassays, the antibody binding sites, antibody specificity and affinity, the combination and orientation of the capture and detection antibody as well as the assay composition, use of buffers, choice of readout systems, and calibration play an enormous role for the results obtained and the potential impact of influencing factors. While our results show a good preanalytical robustness and reproducibility of the specific test systems used, this must be investigated for other PD marker assays again. Second, it must be emphasized that soluble PD markers are not a homogenous group but comprise those that are freely circulating, protein-bound, and vesicle-associated in the blood plasma and serum. Different centrifugation speeds that are feasible in routine laboratories were investigated in this study but no relevant differences in measured marker concentrations were found. Further pre-treatment steps such as filtration, ultracentrifugation, and more could have been tested too, however, this would not be feasible in routine lab procedures. Third, weighing between an analysis considering many different factors and statistical sound sample numbers, it was chosen to start with a pilot approach in a small number of samples. Considering that the number of samples would be limited, an elaborated pre-centrifugation, centrifugation, and post-centrifugation design with plenty of time and temperature and sample handling conditions was chosen. Nevertheless, in the case of gross deviations of the results it should have been seen even in this small set of samples. Having identified the interesting factors, these can be investigated in a larger number of samples to confirm the findings. Fourth, the small size of samples included allows only statements on the cohort of healthy patients whereas different circumstances, such as disease status, might influence preanalytical biomarker stability. The PD-L1 biomarker concentrations were very low in healthy participants and resulted in higher percentages of variation. It is so far unclear whether this will be of clinical relevance. All these factors must be considered now proceeding to the measurement of other clinical cohorts. With more data gathered, new insights on the transferability of results will be revealed. Even though the first measured clinical cohorts will be different cancer types, the biomarkers might also be of considerable interest in a broader scope of indications including cardiovascular disease, infectious diseases such as COVID-19, allergy, and autoimmunity.

## 5. Conclusions

The three soluble biomarkers PD-1, PD-L1, and PD-L2 revealed good preanalytical stability in the presented experiments. This enables the application of all three assays in oncoimmunology research and sensitive monitoring of anti-cancer immunotherapy.

## Figures and Tables

**Figure 1 biomedicines-10-02534-f001:**
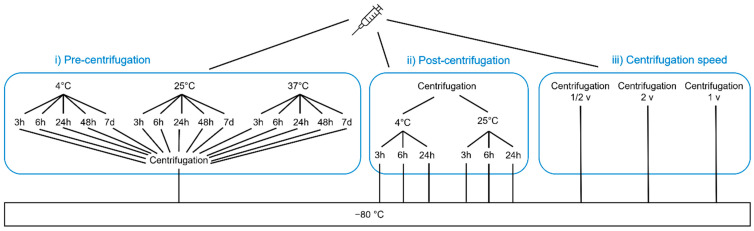
Workflow of the short-term stability experiment. The chart summarizes all the different conditions tested in the short-term stability experiments. The blood is collected all at the same time and split into aliquots for downstream processing. The left side of the graphic (i) includes all temperature and time points before centrifugation, whereas those regarding after centrifugation conditions can be found in the middle (ii). The variation in centrifugation speed is positioned on the right. The control sample, centrifuged and frozen immediately is located on the right end of the figure. v, velocity.

**Figure 2 biomedicines-10-02534-f002:**
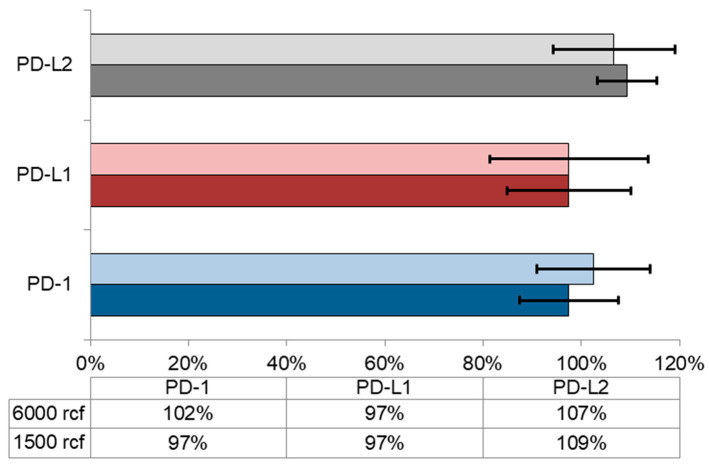
Influence of the variation of centrifugation speed. Mean recoveries obtained after a 10 min centrifugation at 1500 rcf (dark) and 6000 rcf (light) are compared with the reference speed of 3000 rcf in a bar chart. Coefficients of correlation are depicted in the form of error bars.

**Figure 3 biomedicines-10-02534-f003:**
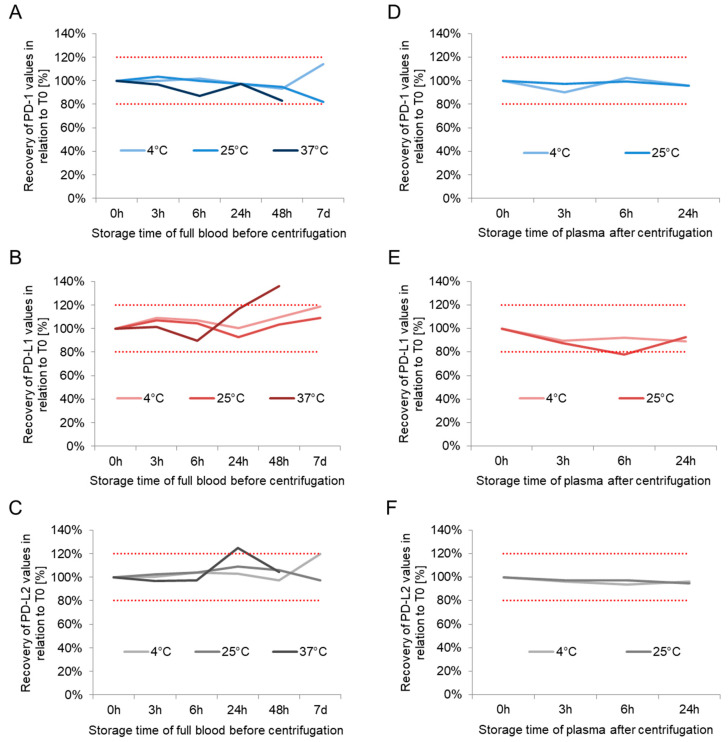
Short-term stability results before and after centrifugation. Mean recoveries obtained in the short-term stability experiment were measured in nine different heparin plasma samples. PD 1, PD L1, and PD L2 results of the whole blood samples are visualized in the left part of the figure (**A**,**C**,**E**) and those of the plasma in the right part (**B**,**D**,**F**). For clarity, coefficients of variations are provided in Table 1 The blue lines represent before centrifugation results meanwhile the pink ones encompass all after centrifugation results. Dotted red lines mark the 80% and 120% borders.

**Figure 4 biomedicines-10-02534-f004:**
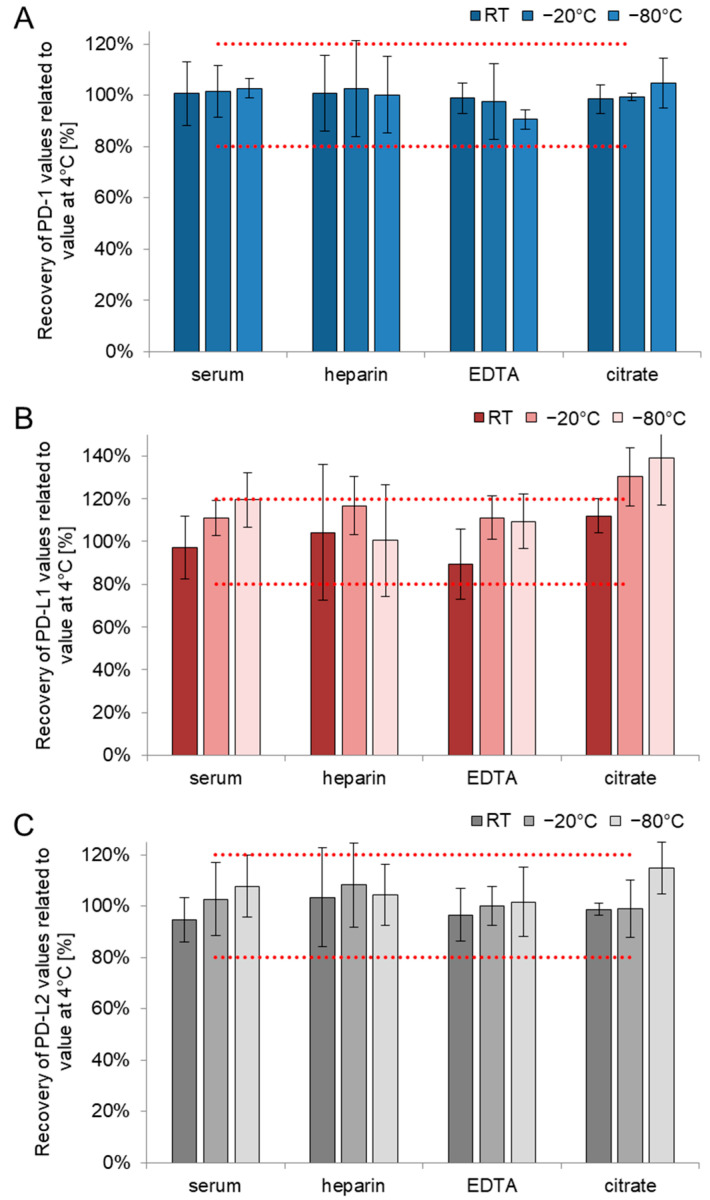
Freeze recoveries in four different blood sampling tubes for PD-1 (**A**), PD-L1 (**B**) and PD-L2 (**C**). The 3 h storage at 4 °C is compared to room temperature, −20 °C, and −80 °C in the four popular blood sampling tubes (serum, heparin, EDTA, and citrate). The figure presents the mean recoveries (*n* = 5) for all three biomarkers. Dotted red lines encompass the range between 80% and 120%. RT, room temperature.

**Figure 5 biomedicines-10-02534-f005:**
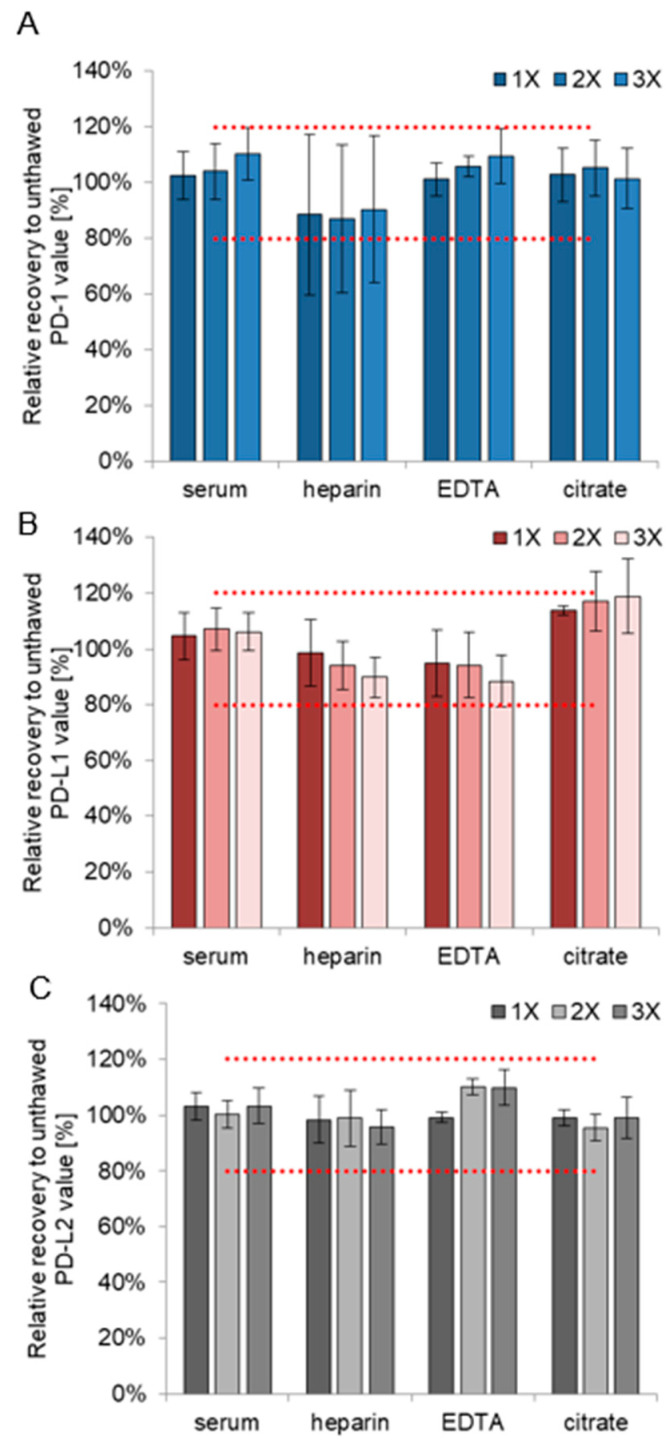
Recoveries after up to three freeze–thaw cycles for PD-1 (**A**), PD-L1 (**B**) and PD-L2 (**C**). The figure compares unthawed to once, twice, and three times refrozen sample values. The samples are taken from five patients in four different blood sampling tubes (serum, heparin, EDTA, and citrate). Dotted red lines encompass the range between 80% and 120%.

**Table 1 biomedicines-10-02534-t001:** Mean recoveries calculated from heparin plasma in course of the short-term stability experiment.

Assay	Condition	Time				
		3 h	6 h	24 h	48 h	7 days
PD-1	BC 4 °C	100% (9%)	102% (12%)	97% (11%)	93% (17%)	114% (10%)
	BC 25 °C	104% (11%)	100% (10%)	97% (7%)	95% (15%)	84% (34%)
	BC 37 °C	97% (9%)	87% (12%)	97% (17%)	83% (26%)	-
	AC 4 °C	90% (7%)	102 (23%)	96% (7%)	-	-
	AC 25 °C	97% (14%)	99% (10%)	96% (12%)	-	-
PD-L1	BC 4 °C	109% (27%)	107 (28%)	100% (32%)	110% (19%)	119% (17%)
	BC 25 °C	107% (17%)	104% (16%)	92% (22%)	103% (18%)	109% (13%)
	BC 37 °C	101% (28%)	90% (25%)	117% (16%)	103% (18%)	-
	AC 4 °C	89% (17%)	92% (31%)	89% (26%)	-	-
	AC 25 °C	88% (25%)	78% (29%)	93% (28%)	-	-
PD-L2	BC 4 °C	100% (8%)	104% (9%)	103 (14%)	97% (28%)	120% (16%)
	BC 25 °C	102% (6%)	104% (16%)	109% (14%)	106% (17%)	97% (35%)
	BC 37 °C	97% (16%)	97% (21%)	124% (15%)	104% (27%)	-
	AC 4 °C	96% (14%)	94% (19%)	96% (9%)	-	-
	AC 25 °C	97% (14%)	97% (19%)	95% (17%)	-	-

BC: before centrifugation; AC: after centrifugation; mean recovery, coefficient of variation in hyphens.

**Table 2 biomedicines-10-02534-t002:** Final selected conditions for the standard operating procedure for PD-1, PD-L1, and PD-L2.

	PD-1	PD-L1	PD-L2
Material	Serum, Heparin, EDTA, Citrate	Serum, Heparin, EDTA	Serum, Heparin, EDTA, Citrate
Time whole blood			
4 °C	Up to 7 d	Up to 7 d	Up to 7 d
25 °C	Up to 7 d, 3 h	Up to 7 d, 3 h	Up to 7 d, 3 h
37 °C	Up to 7 d	Up to 24 h	Up to 6 h
Centrifugation	1500,3000,6000 rcf	1500,3000,6000 rcf	1500,3000,6000 rcf
Time after plasma			
4 °C	Up to 24 h, 3 h	Up to 24 h, 3 h	Up to 24 h, 3 h
25 °C	Up to 24 h, 1 h	Up to 3 h, 1 h	Up to 24 h, 1 h
Storage tubes	Cryotube	Sample tube, cryotube	Cryotube
Plate application	Mixing (pipette or vortexer), centrifugation	Mixing (pipette or vortexer), centrifugation	Mixing (pipette), centrifugation
Freezing process			
Serum	Stable, −80 °C	Stable, −80 °C	Stable, −80 °C
Heparin plasma	Stable, −80 °C	Stable, −80 °C	Stable, −80 °C
EDTA plasma	Stable, −80 °C	Stable, −80 °C	Stable, −80 °C
Citrate plasma	Stable, −80 °C	Not stable	Stable, −80 °C
Freeze–thaw cycles			
Serum	3	3	3
Heparin plasma	1	1	3
EDTA plasma	3	3	3
Citrate plasma	3	Not stable	3

d: day(s), h: hour(s), bold: favored conditions.

## Data Availability

The data will be made available by the authors on request.

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
