# Peer review of "Robust Preanalytical Performance of Soluble PD-1, PD-L1 and PD-L2 Assessed by Sensitive ELISAs in Blood"

_biomedicines, 2022, doi:10.3390/biomedicines10102534_

Round 1

Reviewer 1 Report

in the attached file

Author Response

See attached PDF

Reviewer 2 Report

The topic of the results article is important from the point of view of setting standards for further research and comparability of published results.

I have several comments on the evaluated manuscript.

Abstract:

The article evaluates the stability of proteins, however, the abstract does not contain the word protein even once. Please include it in the appropriate place of the abstract.

The abstract should contain the number of tested samples or biological replicates. In short, information about the size or scope of testing.

Line 27-28 Sample values  - specify the units

Text of the manuscript:

Line 43 Provide a citation.

Line 44 Add at least one additional citation.

Line 48 PD markers - describe the abbreviation

Line 67 a variety of clinically ---  add: short-term stability, freeze effects, refreeze effects and sample preparation before application

Line 88 low, medium and high concentration - please specify

Line 280 CLSI-guideline - describe the abbreviation

Line 291 acceptable

acceptable recoveries for PD-1 between 80 and 120% - presented only in the abstract, state here or in a more appropriate place in the text, what is acceptable

Author Response

See attached PDF
